# Effect of Electrostatic Interactions on the Interfacial Energy between Thermoplastic Polymers and Graphene Oxide: A Molecular Dynamics Study

**DOI:** 10.3390/polym14132579

**Published:** 2022-06-25

**Authors:** Mayu Morita, Yutaka Oya, Nobuhiko Kato, Kazuki Mori, Jun Koyanagi

**Affiliations:** 1Department of Materials Science and Technology, Graduate School, Tokyo University of Science, Tokyo 125-8585, Japan; 8218081@ed.tus.ac.jp; 2Research Institute for Science & Technology, Tokyo University of Science, 6-3-1 Niijuku, Katsushika-ku, Tokyo 125-8585, Japan; 3Science and Engineering Systems Division ITOCHU Techno-Solutions Corporation, Minato-ku, Tokyo 105-6950, Japan; nobuhiko.kato@ctc-g.co.jp (N.K.); kazuki.mori.013@ctc-g.co.jp (K.M.); 4Department of Materials Science and Technology, Tokyo University of Science, Tokyo 125-8585, Japan; koyanagi@rs.tus.ac.jp

**Keywords:** interfacial energy, molecular dynamics, thermoplastic resin, carbon fiber, composite

## Abstract

In this study, the atomistic-scale mechanisms affecting the interfacial stability of a thermoplastic polymer/graphene oxide interface are investigated using molecular dynamics simulations. Different combinations of thermoplastic polymers (polyethersulfone (PES) and polyetherimide (PEI)) and graphene oxides modified with –O–, –OH, and –COOH are prepared. PES is found to be more strongly stabilized with modified/functionalized graphene oxide in the order of –COOH, –OH, –O–, which is opposite to the stability order of PEI. Our results suggest that these orders of stability are governed by a balance between the following two factors resulting from electrostatic interactions: (1) atoms with a strong charge bias attract each other, thereby stabilizing the interface; (2) the excluded-volume effect of the functional groups on graphene oxide destabilizes the interface by preventing π-π stacking of aromatic rings.

## 1. Introduction

Recently, carbon fiber-reinforced thermoplastic polymer composites (CFRTPs) have attracted significant attention for application as structural materials in automobiles [1,2,3,4,5,6,7,8,9]. A CFRTP generally exhibits superior toughness, productivity, and recyclability; however, it demonstrates inferior specific rigidity and strength relative to carbon-fiber-reinforced plastic composites (CFRPs), which are usually employed in large transport. These differences between the properties of CFRTP and CFRP originate from differences in their matrix polymers. A CFRTP consists of mutually entangled thermoplastic polymers that can be molded at a temperature above the glass transition temperature. However, the matrix part of a CFRP constitutes thermosetting polymers that form covalently crosslinked network structures, which induce higher specific rigidity and strength but poor recyclability. With increasing demands to reduce the environmental impact, the development of CFRTPs with superior thermomechanical properties has become increasingly important.

The performance of CFRTPs not only depends on the properties of reinforcing fibers and matrix polymers but also on their interfacial properties [10,11,12,13,14]. The force applied to the composite material is transmitted through the interface between the fiber reinforcement and polymers, and the composite material easily breaks when the interfacial adhesiveness is weak [6,15,16]. Various studies have been conducted to improve the interfacial strength [17,18]. These studies can be roughly classified into two groups depending on the approach adopted to improve the interfacial adhesiveness [19]. One approach is to modify the surface of the carbon fiber by functionalization via oxidation treatment or electroplating. The other is to add another type of carbon-based material, such as carbon nanotubes, to the matrix [20]. In this study, we focused on the former approach; that is, we investigated the influence of carbon fiber functionalization on the interfacial energy at the atomistic scale using full atomistic molecular dynamics (MD) simulations.

Various simulation studies have been conducted to clarify the thermomechanical properties of polymers [21,22,23,24], reinforcements [25,26], and composites [27,28,29,30,31,32,33,34,35,36,37,38,39,40,41,42,43,44] using MD simulations. For composites of thermosetting polymers and carbon-nanotubes (CNT), Alien et al. and Park et al. conducted pullout tests to investigate the effect of the atomistic-scale mechanism on the interfacial shear modulus [27,28]. For thermosetting polymer and graphene systems, Mahmud et al. simulated the effects of force fields and the formation of polymer cross-links on the interface energy between the polymer and graphene [29]. Sun et al. investigated the influence of graphene on the diffusion coefficient, density of epoxy, and glass transition temperature [30]. Salahshoor and Rahbar analyzed the nanoscale interfacial fracture toughness between graphene and epoxy by accurately modeling the atomistic behavior during the curing process [31]. Oya et al. simulated a laminated graphene oxide; wherein intergraphene layers were connected through thermosetting polymers. They revealed that both covalent and hydrogen bonding have an effect on Young’s modulus and strength [32]. For a system consisting of thermoplastic polymers and CNTs, Shen et al. reproduced polymer-grafted nanotubes to determine the viscoelastic behavior using a coarse-grained MD model [33]. Yang et al. investigated the correlation between polymer conformations around a CNT and the interfacial energy [34]. Eslami and Behronz investigated the effect of surface curvature on the structure and dynamic behavior of thermoplastic polymers [35]. For a thermosetting polymer and graphene system, which is similar to the system used in this study, Lee et al. investigated the orientation of polystyrene around the interface [36]. Lee et al. simulated a pullout test of graphene from a polymer matrix and reproduced the stress-strain behavior [37]. Equilibrium (structural and energic) and dynamic properties were systematically investigated by Alian et al. [38]. Recently, there have been studies focusing on electrostatic interactions at the polymer-carbon fiber interface, including the following studies. Duan et al. investigated the interface cohesive energy between CNT/epoxy nanocomposites using coarse-grained (CG) MD simulation [39]. They found that CGMD can reproduce the interfacial properties by adjusting the parameter of Lennard–Jones potential and the degree of course-graining. Moghimikherabadi et al. showed the electrostatic interaction improves the stiffness and toughness of ionic polymer nanocomposites using CGMD [40]. Karatasos and Kritikos performed a full-atomic MD simulation for the graphene oxide/poly(acrylic acid)nanocomposite [41]. They found that the dispersion of the GO flakes is characterized by the formation of oligomeric clusters. They also found that hydrogen bonding affects the dynamics of the polymers. Zhang et al. studied nanoscale toughening of ultrathin graphene oxide-polymer composite, and their results indicated that both hydrogen bonding and van der Waals interactions significantly affect the toughness by preventing the graphene-bound polymers from fracturing [42].

As indicated by the aforementioned studies, the MD simulation used in this study has certain advantages over the finite element method (FEM), which is usually used to investigate the mechanical properties of a composite. MD simulations calculate the thermodynamic properties based on atomic motion, whereas FEM requires the constitutive law of a material expressed as a continuum body. Therefore, MD simulations enable us to investigate the dependence of the interfacial strength on the material species at the atomistic scale [43]. The purpose of this study was to determine the atomistic factors that aid in selecting an appropriate composite with excellent interfacial strength by using MD simulations. For this purpose, we selected polyethersulfone (PES) and polyetherimide (PEI) as matrix polymers [44,45] and graphene sheets with and without functionalization as reinforcements. Furthermore, for the functionalized graphene, we selected –O–, –COOH, and –OH as functional groups bonded to the carbon atoms on the graphene sheet. The interfacial strength is strongly correlated to the interfacial stability; that is, the strength appears to be higher when the interfacial energy is lower. Therefore, an MD simulation was adopted to obtain the interfacial energies of eight species of the composite model, which constituted combinations of two types of matrix polymers and four types of graphene.

The remainder of this paper is organized as follows. The next section outlines the procedures used to construct the composite model and its equilibrium structures using MD simulations. In the third section, the simulation results and corresponding discussions are presented. Finally, concluding statements are provided in the fourth section of this manuscript.

## 2. Method

### 2.1. Creating Composite Models

MD simulations were employed to calculate the energy at the interface between the graphene sheets and polymers. For this purpose, the layered structure of the graphene sheets and polymers was constructed using the following procedures.

First, four types of graphene sheets and two types of matrix polymers were fabricated separately. Atomistic models of a pure graphene sheet and three types of functionalized graphene sheets modified with –OH, –O–, and –COOH were prepared using Graphene Builder in PolyPerGen [46,47]. Each functional group was bonded to a randomly selected carbon atom on one side of the graphene sheet. The number ratio of functional groups to carbon atoms was set to 0.1. All graphene sheets were 60 Å × 61 Å in the same area. The MD models of the pure and functionalized graphene sheets are presented in Figure 1. For matrix polymers, models of PES ((C_12_H_8_O_3_S)_n_, degree of polymerization *n* = 10) and PEI ((C_37_H_24_O_6_N_2_)_n_, degree of polymerization *n* = 4) were constructed using Marvin Sketch [48] and PolyPerGen. The atomistic models of PES and PEI after structural optimization are presented in Figure 2. In our simulation, the all-atom optimized potentials for liquid simulation (OPLS-AA) force field was employed to reproduce the molecular structure [49], and the particle mech Ewald method (PME) was used for computing Coulomb interactions. Cutoffs of Lennard–Jones (LJ) and Coulomb interactions are both 1.0 nm. All simulations were performed using the software GROMACS [50]. The electrostatic potential charge obtained by density functional calculations using B3LYP/6-31G (Hamiltonian/basis set) was set for each atom to reproduce the electrostatic field around each molecule [51,52].

Next, three graphene layers were placed at the center of the simulation cell with interlayer distances of 0.335 nm. Only pure graphene was used as the second layer, whereas pure or functionalized graphene sheets were used as the first and third layers, wherein the side containing functional groups faced outward. For the matrix portion of the composite, PESs or PEIs were randomly arranged above and below the graphene layers such that the density of atoms in the simulation cell was about 0.3 g/cc. It indicates the system is stretched in the z-direction compared to the equilibrium state; here, the z-axis direction is defined as the direction perpendicular to the graphene plane. The total number of atoms in the simulation cell was approximately 30,000.

### 2.2. Relaxation Calculation

The relaxation calculation procedure used to obtain the equilibrium structure of the composite models is described below.

First, structural optimization was performed to minimize the internal energy of the system while maintaining the volume of the system at a constant value.

Next, the equilibrium structure at 600 K was obtained by stepwise relaxations under *NVT* and *NPT* ensembles for 200 ps and 1 ns, respectively, where *N* denotes the total number of atoms, *V* denotes the system volume, *P* denotes the pressure, and *T* denotes the temperature. In the relaxation under an *NPT* ensemble, a pressure of 1 bar was applied in the XY-plane directions, and a pressure of 10 bar was applied along the z-axis direction to bring a lower density system closer to a system with appropriate density.

Finally, the equilibrium structure at room temperature (*T* = 300 K) was obtained by stepwise relaxation as follows: In the first step, the temperature of the system was decreased from 600 K to 300 K at a cooling rate of 6.0 × 10^9^ K/s under an *NVT* ensemble. Note that this cooling rate is much greater than that in the usual experiment (tens K/hour). The effect of the different conditions on the structure eventually disappears with sufficiently long relaxation calculations. Next, a relaxation calculation was performed for 200 ps under an *NVT* ensemble at *T* = 300 K. The equilibrium structure was obtained by relaxation for 10 ns under an *NPT* ensemble at *T* = 300 K and *P* = 1 bar along with the xyz-axis directions. The equilibrium structures of the composite model composed of PEI and graphene with -OH are illustrated in Figure 3. Using the equilibrium composite model, the interfacial energy per unit area, $E_{\mathrm{interface}}$, was calculated using the following equation:(1)$$E_{\mathrm{interface}}=\frac{E_{\mathrm{total}}-\left(E_{\mathrm{polymer}}+E_{\mathrm{graphene}}\right)}{2A}$$
where $E_{\mathrm{total}}$ represents the internal energy of the entire composite system, $E_{\mathrm{polymer}}$ denotes the internal energy of the polymers, $E_{\mathrm{graphene}}$ indicates the internal energy of the graphene sheets, and A denotes the interfacial area between the graphene sheets and polymers.

## 3. Results and Discussion

Calculation results of the interfacial energy are presented in Table 1. The interfacial energies are all negative, indicating that the interaction between the polymer and graphene stabilizes the system better compared with the polymer and graphene alone. Compared with PEI, PES better stabilizes the interface with functionalized graphene. The stability of the interface between PES and functionalized graphene decreases in the order of –COOH, –OH, and –O, which is in direct contrast to the stabilization order for PEI. One of the primary factors influencing interfacial stability is the electrostatic interaction between the polymer and graphene. We investigated the basis for the change in interfacial energy by considering the charge distribution in each polymer chain as follows.

The primary differences between these polymer structures are their constituent atoms, where SO_2_ and N are connected to the unsaturated ring structures in PES and PEI, respectively. The charge distribution around these atoms affects the electrostatic interactions at the interface. Figure 4 and Figure 5 illustrate charge distributions in the monomers for PEI and PES, respectively. In these figures, it can be observed that the N atom of PEI carries a charge of −0.754*e* and the S atom of PES carries a charge of +1.374*e*, which represents the largest charge bias in each monomer. The charge bias of each atom is known to be influenced by the electronegativity $\left(\chi \right)$ of its neighboring atoms. Electrons of the S atom $\left(\chi =2.5\right)$ of PES are attracted to the O atoms $\left(\chi =3.5\right)$ inducing a positive charge bias of the S atom and a negative charge bias of the O atom. In contrast, the N atom $\left(\chi =3.0\right)$ of PEI attracts electrons from neighboring C atoms $\left(\chi =2.5\right)$ resulting in a negative charge bias of the N atom. These atoms of PES exhibit a greater charge bias than those of PEI because the charge bias is essentially proportional to the electronegativity between neighboring atoms.

Figure 6 illustrates the charge distributions in a repeating unit of pure graphene and in a region around the functional group of functionalized graphene. In these figures, the charge bias of functionalized graphene is greater than those of pure graphene. For example, the O atom demonstrates the largest charge bias for all functional groups, resulting in a value of −0.556*e* for –COOH, −0.550*e* for –OH, and −0.281*e* for –O. In the following discussion, pure graphene and functionalized graphene will be referred to as “GR” and “FGR”, respectively. Furthermore, functionalized graphene modified with –O–, –COOH, and –OH will be referred to as “FGR–O,” “FGR–COOH,” and “FGR–OH”, respectively.

To investigate the influence of the charge bias on the atomistic structure around the interface, the radial distribution functions (RDFs) between the functional groups in functional graphene and each atom component in the polymer are presented in Figure 7. Each curve in the RDF is selected for a combination of atoms with positive and negative charge biases that are electrostatically attracted to each other, namely H atom in FGR and O atom in PEI or PES (red curve), O atom in FGR and H atom in PEI or PES (blue curve), O atom in FGR and S atom in PES (yellow curve), and H atom in FGR–O and N atom in PEI (green curve). In these figures, the horizontal axis represents the distance from the O or H atom in FGR. Figure 7a,b depict the RDFs for FGR–OH and FGR–COOH, respectively, and the upper and lower figures provide the corresponding data for PEI and PES, respectively. Both RDFs for PEI and PES commonly exhibit two peaks: O atom at 0.20 nm and H atom at 0.25 nm. Furthermore, the N atoms in PEI and S atoms in PES demonstrate peaks at the same distance of 0.40 nm. These peaks indicate the electrostatic attractive interactions between the polymer and FGR. In particular, the blue curves, indicating RDFs between the O atom in the polymers and H atom in FGR–OH or FGR–COOH, exhibit a stronger peak near the interface, which indicates the existence of hydrogen bonds. PES is more affected by electrostatic interactions than PEI because PES has a sharper RDF peak owing to its stronger charge bias. Therefore, the interface of PES with FGR is expected to be more stable than that of PEI, as indicated in Table 1. In particular, for PES, the magnitude of the interfacial energy is governed by electrostatic interactions with the functional groups of FGR. For example, the combination of PES and FGR–COOH consists of atoms with the strongest charge biases, resulting in the lowest interfacial energy.

In the case of PEI, the trend of charge observed in the magnitude of interfacial energy is opposite to that observed for the PES. For example, the interfacial energy is the highest with FGR–COOH (i.e., unstable interface), although the corresponding RDFs demonstrate sharp peaks owing to electrostatic interactions between the functional groups, as mentioned above. For PEI, the interfacial energy is the lowest with FGR–O despite the absence of noticeable peaks in the RDF compared to other combinations, as shown in Figure 7c. Thus, the stability of the PEI interface cannot only be explained based on attractive interactions with the functional groups of FGR. Previous studies have suggested that aromatic rings between graphene and a polymer stack parallel to each other [53,54]. This is referred to as π-π stacking via electrostatic interactions caused by un-localized electrons around the ring. Figure 8 presents the density profile of the atoms in PEI and PES with respect to the position of the z-axis, where z = 0 represents the position of the second graphene layer. Sharp peaks are observed around z = 0.7 and −0.7 for pure graphene. These peaks indicate the existence of π-π stacking. The amplitude of the PEI peak is larger than that of the PES peak, and PEI forms a more stable interface with pure graphene via π-π stacking. Figure 8 also indicates that the amplitude of these peaks formed due to π-π stacking decreases when the functional groups are bonded to the graphene. Therefore, π-π stacking is prevented by the functional groups in FGR, inducing a higher energy at the interface. In particular, in the case of –COOH, the polymer density slightly changes around the interface. The –COOH group on FGR exhibits a large excluded volume, which prevents the aromatic ring of PEI from approaching graphene. However, –O– on FGR cannot rotate, and its excluded volume is smaller, which does not strongly inhibit π-π stacking.

The interfacial stabilities of the different composite models are presented in Figure 9. The stability is quantitatively defined as the absolute value of the contribution of energy to the interfacial energy. This stability can be divided into two types of contributions: the Coulomb potential and the Lennard–Jones (LJ) potential. The stability based on the Coulomb potential is related to the attractive interaction between atoms resulting from a large charge bias, whereas the stability based on the LJ potential is attributed to π-π stacking between the aromatic rings. These figures demonstrate the existence of a trade-off relationship between attractive interactions and π-π stacking; this trade-off relationship emerges owing to the fact that their magnitudes are inversely related. These figures also support the arguments described thus far. By comparing Figure 9a,b, it can be concluded that both PEI and PES stabilize the interface with pure graphene due to π-π stacking. However, in the case of FGR–COOH and FGR–OH, the interface is stabilized by local attractive interactions as opposed to π-π stacking, resulting in the most stable interface from the combination of PES and FGR–COOH. PEI forms a stable interface with FGR–O because both effects are balanced.

In conclusion, there exist two mechanisms via which the functional groups introduced into graphene sheets contribute to interfacial stability. First, a functional group with a larger charge bias attracts the atoms in the polymer, which stabilizes the interface. Second, a functional group with a large excluded volume prevents the formation of π-π stacking structures, destabilizing the interface. Therefore, it is crucial to select an appropriate combination of functional groups on FGR and polymers by considering the aforementioned trade-off relationship.

## 4. Conclusions

Atomistic-scale mechanisms that determine the interfacial stability between thermoplastic polymers and graphene oxides are important for the development of CFRTPs with excellent mechanical properties. In this study, the interfacial energy was evaluated using full atomistic MD simulations. We prepared eight species of layered structures of CFRTPs by combining two types of thermoplastic polymers, namely PES and PEI, and four types of graphene sheets modified with and without different functional groups, including –O–, –OH, and –COOH. Based on the MD simulation results, the stability of the interface between PES and functionalized graphene decreased in the order of –COOH, –OH, and –O. This is in contrast to the interface stability with PEI, which decreased in the order of –O–, –OH, and –COOH. RDF analysis suggested that electrostatic interactions between atoms with a stronger charge bias around the interface play an important role in determining the order of the interfacial stability of PES. The interface between PES and graphene-modified with –COOH was found to be the most stable among the various species investigated. This is attributed to the attractive forces between the atomic combinations of S and H atoms in PES and O atoms in FGR–COOH, which have positive and negative charge biases, respectively. Furthermore, our results also suggest that the functional groups of FGR prevent π-π stacking of aromatic rings between FGR and the polymer via excluded-volume effects. Therefore, the functional groups on graphene sheets play an important role in stabilizing and destabilizing the interface. Our findings may be beneficial for the selection of appropriate materials in the development of CFRTPs with superior thermomechanical properties.

## Figures and Tables

**Figure 1 polymers-14-02579-f001:**
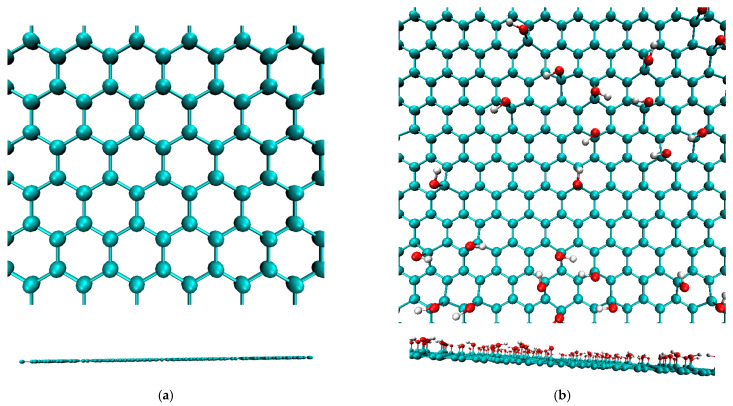
Snapshots of graphene sheets with and without functionalized groups. The upper and lower figures represent the top and side views, respectively. (**a**) Without functional group. (**b**) With functional group –OH. (**c**) With functional group –COOH. (**d**) With functional group –O–.

**Figure 2 polymers-14-02579-f002:**
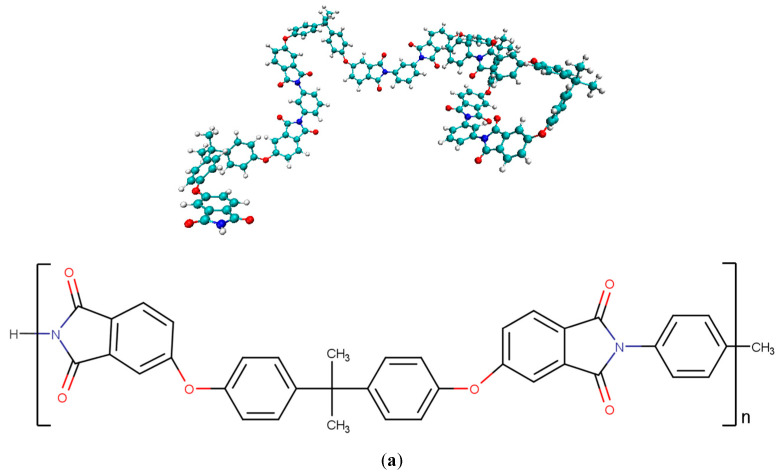
Snapshots and chemical structures of thermoplastic polymers: (**a**) PEI and (**b**) PES.

**Figure 3 polymers-14-02579-f003:**
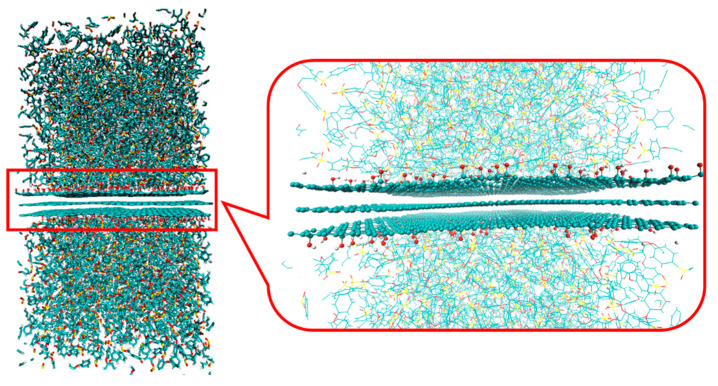
Snapshot of the composite model of PEI and graphene with –OH in the equilibrium state. The enlarged interface region and graphene sheets are highlighted in the right figure.

**Figure 4 polymers-14-02579-f004:**
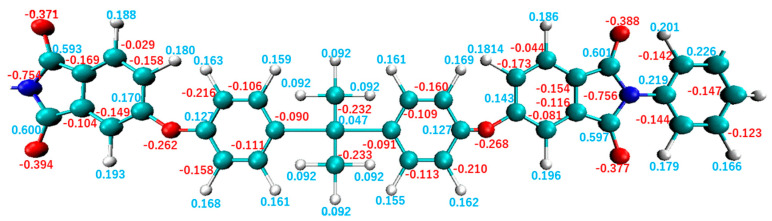
Charge distribution in a monomer of PEI.

**Figure 5 polymers-14-02579-f005:**
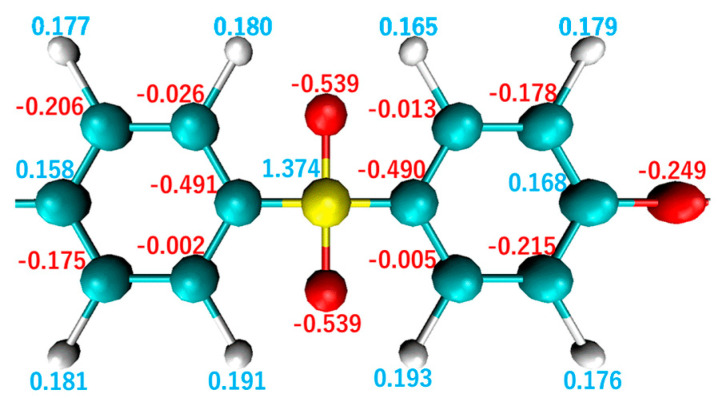
Charge distribution in a monomer of PES.

**Figure 6 polymers-14-02579-f006:**
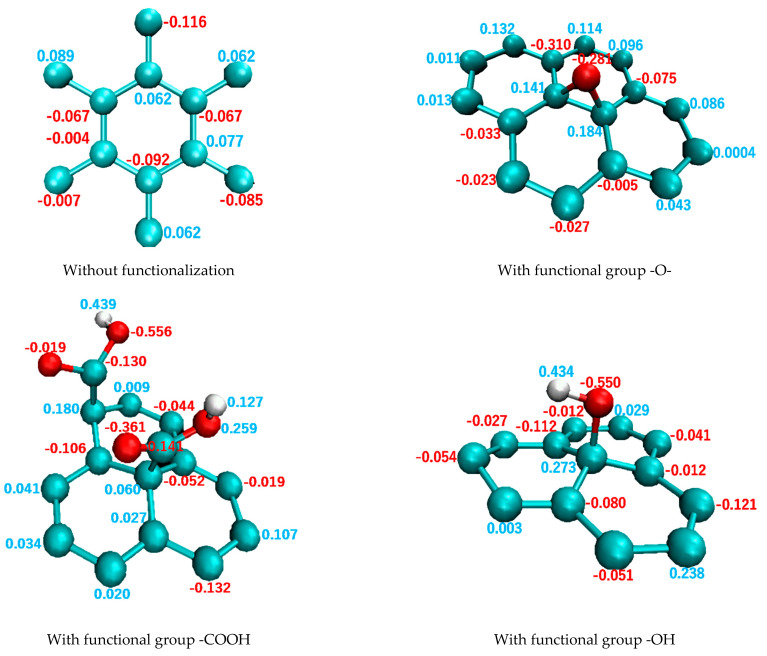
Charge distributions in a repeating unit of graphene and a region around the functional group of functionalized graphene.

**Figure 7 polymers-14-02579-f007:**
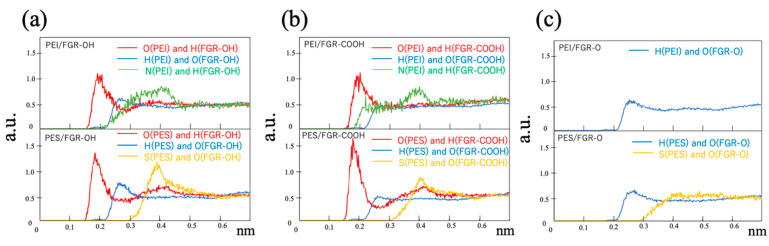
RDFs for atom pairs with positive and negative charge biases. (**a**) Functionalized graphene (FGR) modified with –OH, (**b**) FGR modified with –COOH, and (**c**) FGR modified with –O–. The upper and lower figures provide the corresponding data for PEI and PES, respectively.

**Figure 8 polymers-14-02579-f008:**
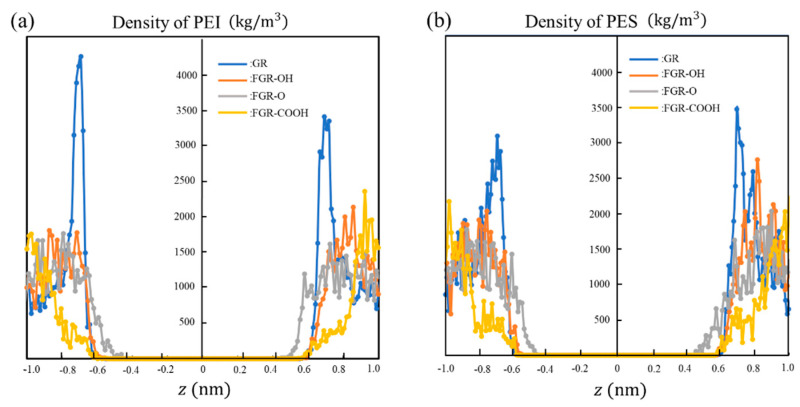
Density profiles of atoms in (**a**) PEI and (**b**) PES for composite models with different graphene oxides. The horizontal axis represents the position of the z-axis, which is perpendicular to the graphene interface.

**Figure 9 polymers-14-02579-f009:**
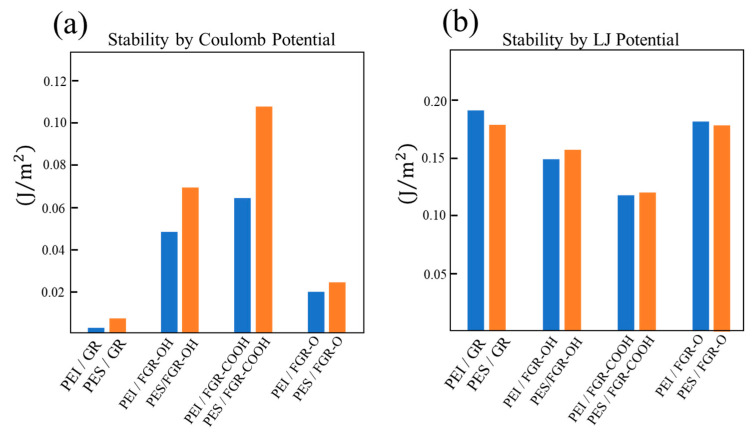
Interfacial stability is based on the (**a**) Coulomb potential and (**b**) Lennard–Jones potential for different composite models. The stability is defined by the absolute values of potential energies.

**Table 1 polymers-14-02579-t001:** Calculation results of the interfacial energy $\left(J/m^{2}\right)$. The upper and lower rows represent the mean value and the error, respectively.

	Functional Groups on Graphene			
	None	–OH	–COOH	–O–
PEI	−0.194 (±1.72 × 10^−3^)	−0.197 (±2.85 × 10^−3^)	−0.182 (±1.99 × 10^−3^)	−0.201 (±1.68 × 10^−3^)
PES	−0.186 (±3.83 × 10^−3^)	−0.226 (±0.51 × 10^−3^)	−0.228 (±0.76 × 10^−3^)	−0.202 (±0.11 × 10^−3^)

## Data Availability

Not applicable.

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
