# Peer review of "Effect of Electrostatic Interactions on the Interfacial Energy between Thermoplastic Polymers and Graphene Oxide: A Molecular Dynamics Study"

_polymers, 2022, doi:10.3390/polym14132579_

Round 1

Reviewer 1 Report

In this manuscript, Morita et al. study the adhesion of thermoplastic energies to (functionalized) graphene oxide surfaces via all-atom molecular dynamics (MD) simulations. They focus on two different polymers, i.e., PES and PEI, and four different substrate types, finding that the stability of the films strongly depend on both electrostatic and van der Waals interactions. Overall, the studies have been carried out carefully and the results are presented in a clear manner. I have only few suggestions, which should be addressed before I can recommend publication of this manuscript:

- The authors apply normal and lateral pressures between 1 and 10 bar during their simulation, but do not explain why they chose these specific values. Are these pressures representative of typical experimental conditions?

- The cooling rate in the simulations is (unrealistically) high at 6x10^9 K/s. I understand that this choice is a direct limitation of the employed all-atom model, but the authors should still briefly discuss typical experimental values and whether their fast cooling will lead to simulation artifacts or not.

- The interfacial energy defined in Eq. (1) contains only potential energy terms, but lacks entropic contributions to the interfacial free energy. Such contributions might play an important role, given the potentially different packing entropies between the different polymer/surface combinations. The authors should comment on this point.

- The data shown in Table 1 is lacking error bars, making it difficult to decide whether the small energy differences are statistically significant or not. Therefore, error bars should be added wherever appropriate.

Author Response

Replies to Referee #1

  Thank you very much for your careful reading of our manuscript and for your valuable comments.  According to your criticisms and comments, we made a revision of the manuscript.  Replies to your individual comments are listed below.

[Comment #1]
The authors apply normal and lateral pressures between 1 and 10 bar during their simulation, but do not explain why they chose these specific values. Are these pressures representative of typical experimental conditions?

[Reply]

In the process of achieving equilibrium state, pressures of 10 bar and 1 bar are applied in the z-direction and lateral directions of the system, respectively. One bar is almost the same as room pressure; however, 10 bar is a pressure far removed from the experimental conditions.  

When these pressures are initially applied, the density is much lower than the equilibrium state because the system is stretched in the z-direction.  To bring such a lower density system closer to a system with appropriate density, a pressure of 10 bar is applied in the z-direction. This specific pressure value was determined as appropriate in terms of numerical stability and calculation resource.

Therefore, to clarify the reason for selecting the specific pressure value, we have added the following sentence in the revised manuscript.

(Page 5, line 9 from the bottom)

For the matrix portion of the composite, PESs or PEIs were randomly arranged above and below the graphene layers such that the density of atoms in the simulation cell was about 0.3 g/cc. It indicates the system is stretched in the z-direction compared to the equilibrium state; here, the z-axis direction is defined as the direction perpendicular to the graphene plane. The total number of atoms in the simulation cell was approximately 30,000.

(Page 6, line 6)

In the relaxation under an NPT ensemble, a pressure of 1 bar is applied in the XY-plane directions, and a pressure of 10 bar was applied along the z-axis direction to bring a lower density system closer to a system with appropriate density. This specific pressure value of 10 bar was determined as appropriate in terms of numerical stability and calculation resource.

------------------------------------------------------------------

[Comment #2]
The cooling rate in the simulations is (unrealistically) high at 6x10^9 K/s. I understand that this choice is a direct limitation of the employed all-atom model, but the authors should still briefly discuss typical experimental values and whether their fast cooling will lead to simulation artifacts or not.

[Reply]

In general, the procedures for reproducing the equilibrium state in MD simulation does not correspond to experiments because the spatio-temporal scales that can be handled are quite different from those of molding process of CFRTPs. As the referee commented, the cooling rate in the MD simulation is un-realistic value compared with the cooling rate in usual experiment (tens K/hour).  The effect of the different conditions on the structure eventually disappears with sufficiently long relaxation calculations. Therefore, the difference in the cooling rate does not influence on the final equilibrium structure. 

Therefore, to clear the difference in molding conditions between MD and experiment and the effects of the difference on the equilibrium structure, the following sentences are added to our revised manuscript.

(Page 6, line 12 from the bottom)

Note that this cooling rate is much greater than that in the usual experiment (a few tens K/hour). The effect of the different conditions on the structure eventually disappears with sufficiently long relaxation calculations.

----------------------------------------------------------------------

[Comment #3]
The interfacial energy defined in Eq. (1) contains only potential energy terms, but lacks entropic contributions to the interfacial free energy. Such contributions might play an important role, given the potentially different packing entropies between the different polymer/surface combinations. The authors should comment on this point.

[Reply]

           Thank you for the important suggestion. As the reviewer pointed out, the entropic contribution is not considered in this study although it is assumed to be important for the stability of whole system.  As previous studies reported, the graphene sheet restricts conformations of thermoplastic polymers, leading the reduction of the entropy and the change in the stability of the system [R1]. To investigate such an influence of the graphene on the conformational entropy, it is necessary to extract molecular scale information associated with the conformation such as end-to-end distance of a single polymer chain [R2]. We will study effects of functional groups of graphene oxide on the end-to-end distances of the thermoplastic polymers as our future task.

(reference)

[R1] Sahraei, A. A.; Mokarizadeh, A. H.; George, D.; Rodrigue, D.; Baniassadi, M.; Foroutan, M. Insights into interphase thickness characterization for graphene/epoxy nanocomposites : a molecular dynamics simulation. Phys. Chem. Chem. Phys. 2019 21(36), 19890-19903.

[R2] Murakami, W.; De Nicola, A.; Oya, Y.; Takimoto, J. I., Celino, M.; Kawakatsu, T., Milano, G. Theoretical and computational study of the sphere-to-rod transition of triton X-100 micellar nanoscale aggregations in aqueous solution: Implications for membrane protein solubilization. ACS Appl. Nano Matter. 2021, 4, 5, 4552-4561.

----------------------------------------------------------------------

[Comment #4]
The data shown in Table 1 is lacking error bars, making it difficult to decide whether the small energy differences are statistically significant or not. Therefore, error bars should be added wherever appropriate.

[Reply]

           Thank you for the important suggestion. According to the reviewer’s comment, the error of the interfacial energy is newly added to Table 1.  

Reviewer 2 Report

Please see the comments in the attached file. These would improve further your manuscript

Author Response

Replies to Referee #2

  Thank you very much for your careful reading of our manuscript and for your valuable comments.  According to your criticisms and comments, we made a revision of the manuscript.  Replies to your individual comments are listed below.

[Comment #1]

The authors should refer in the introduction studies that also involve CFRTP with thermoplastics and not only thermosets. In addition, they should expand previous simulation MD efforts discussion (containing both coarse grained and atomistic simulations) addressing electrostatic interactions on the reinforcement of the polymer matrix

[Reply]

Thank you for the useful comments and suggestions on the references. We have referred the previous studies that provides an overview of recent CFRTP progress [R3-R6]. Further, we have added descriptions of the MD studies discussing the electrostatic interactions to introduction of our revised manuscript.  

(Page 3, line 12 from the bottom)

Recently, there have been studied focusing on electrostatic interactions at the polymer-carbon fiber interface, including the following studies. Duan et al. investigated the interface cohesive energy between CNT/epoxy nanocomposites using course-grained (CG) MD simulation [R7]. They found that CGMD can reproduce the interfacial properties by adjusting the parameter of Lennard Johns potential and the degree of course-graining.  Moghimikherabadi et al. showed the electrostatic interaction improves the stiffness and toughness of ionic polymer nanocomposites using CGMD [R8]. Karatasos and Kritikos performed full-atomic MD simulation for the graphene oxide/poly(acrylic acid)nanocomposite [R9]. They found that the dispersion of the GO flakes is characterized by the formation of oligomeric clusters. They also found that the hydrogen bonding affects the dynamics of the polymers. Zhang et al. studied nanoscale toughening of ultrathin graphene oxide-polymer composite, and their results indicated that both hydrogen bonding and van der Waals interactions significantly affect the toughness by preventing the graphene-bound polymers from fracturing [R10].

(reference)

[R3] Alshammari, B. A.; Alsuhybani, M. S.; Almushaikeh, A. M.; Altotaibi, B. M.; Alenad, A. M.; Alqahtani, N. B.; Alharbi, A. G. Comprehensive review of the properties and modifications of carbon fiber-reinforced thermoplastic composites, Polymers, 2021, 13, 2474.

[R4] Yao, S. S.; Jin F. L., Rhee, K. Y.; Hui, D.; Park, S. J. Recent advances in carbon-fiber-reinforced thermoplastic composites: A review, Composites Part B, 2018, 142, 241-250.

[R5] Ning, F.; Cong, W.; Qiu, J.; Wei, J.; Wang, S. Additive manufacturing of carbon fiber reinforced thermoplastic composites using fused deposition modeling, Composites Part B, 2015, 80, 369-378.

[R6] Jiang, B.; Chen, Q.; Yang, J. Advances in joining technology of carbon fiber-reinforced thermoplastic composite materials and aluminum alloys, The International Journal of Advanced Manufucturing Technology, 2020, 110, 2631-2649.

[R7] Duan, K.; Li, L.; Wang, F.; Meng, W.; Hu, Y.; Wang, X. Importance of interface in the coarse-grained model of CNT/epoxy nanocomposites. Nanomaterials, 2019, 9 (10), 1479.

[R8] Moghimikheirabadi, A.; Karatrantos A. V.; Kroger, M; Ionic Polymer Nanocomposites subjected to uniaxial extension: a nonequilibrium molecular dynamics study, Polymer, 2021, 13 (22), 4001.

[R9] Karatasos, K.; Kritikos, G. Characterization of a graphene oxide/poly(acrylic acid) nanocomposite by means of molecular dynamics simulations. RSC Adv.. 2016, 6, 109267.

[R10] Zhang, X.; Nguyen, H., Daly, M.; Nguyen S. T.; Espinosa, H. D. Nanoscale toughning of ultrathin graphene oxide-polymer composites: mechanochemical insights into hydrogen-bonding/ vander Waals interactions, polymer chain alignment, and steric parameters, Nanoscale, 2019, 11, 12305.

----------------------------------------------------------------------

[Comment #2]

Both of these polymer matrices have rigid groups (aromatic rings) in the backbone and may crystallize on the graphene surface due to the strong interfacial strength, thus the polymer dynamics (not only polymer structure) should also be estimated and assessed for different functionalizations. In which case the mobility of such polymer is reduced dramatically near the graphene surface, and do they crystallize?    

From the radial distribution functions and perhaps from experimental studies of the systems could the authors comment regarding the dispersion of the graphene oxide in such matrices?

[Reply]

As reviewer pointed out, the polymer mobility is reduced near the graphene surface.  However, visualization of the conformation of all polymers near the interface showed no crystallization, such as polymer chains folding to form lamella structures with finite thickness. There are two presume reasons why polymers do not crystalize near the interface. First, PEI and PES used in this study are classified as amorphous polymers. Second, the functional groups on the graphene sheet restricts the polymer conformations, preventing it from folding. Since MD investigation for the crystallization near the interface is a challenging issue, we plan to investigate it using crystalline polymers such as PEEK in the future.

The dispersibility of graphene in polymer matrices is positively correlated with the affinity between the polymer and graphene. When strong attractive forces due to electrostatic interaction between graphene and polymers are present, the dispersion of the graphene is also higher. Therefore, as the reviewer suggested, the RDF results in this study suggested that the combination of PES and graphene oxide with -COOH leads to the higher dispersion.

----------------------------------------------------------------------

[Comment #3]

The force field used in the study is the OPLS and should be mentioned explicitly.

Which method is used to calculate the electrostatic forces in the classical MD simulations? Is it the PM Ewald, PPPM or perhaps Reaction field method ? Which is the cut off used for electrostatics and van der Waals interactions? All the simulation details should be written explicitly in the manuscript.

Which software simulation package was used to perform the MD simulation?

[Reply]

Thank you for the important suggestions. We added the descriptions in the simulation details explicitly to our revised manuscript.

(Page, 5 line 20 from the bottom)

In our simulation, the all-atom optimized potentials for liquid simulation (OPLS-AA) force field was employed to reproduce the molecular structure, and particle mech Ewald method (PME) was used for computing Coulomb interactions.  Cutoffs of Lennard-Jones (LJ) and Coulomb interactions are both 1.0 nm.  All simulations were performed using software GROMACS [R11].

(reference)

[R11] GROMACS software, https://www.gromacs.org/.

Round 2

Reviewer 2 Report

The authors improved the manuscript according to the suggestions